# Cancer Cell Biomechanical Properties Accompany Tspan8-Dependent Cutaneous Melanoma Invasion

**DOI:** 10.3390/cancers16040694

**Published:** 2024-02-06

**Authors:** Gaël Runel, Noémie Lopez-Ramirez, Laetitia Barbollat-Boutrand, Muriel Cario, Simon Durand, Maxime Grimont, Manfred Schartl, Stéphane Dalle, Julie Caramel, Julien Chlasta, Ingrid Masse

**Affiliations:** 1Cancer Research Center of Lyon, CNRS UMR5286, Inserm U1052, University of Lyon, University Lyon 1, 69000 Lyon, France; gael.runel@bio-meca.com (G.R.); noemie.lopez-ramirez@univ-lyon1.fr (N.L.-R.);; 2BioMeca, 60F, Bioserra 2, Av. Rockefeller, 69008 Lyon, France; 3National Reference Center for Rare Skin Disease, Department of Dermatology, University Hospital, INSERM 1035, 33000 Bordeaux, France; 4AquiDerm, University Bordeaux, 33076 Bordeaux, France; 5Developmental Biochemistry, University of Würzburg, 97074 Würzburg, Germany; 6Xiphophorus Genetic Stock Center, Texas State University, San Marcos, TX 78666, USA; 7Dermatology Department, Hôpital Universitaire Lyon Sud, Hospices Civils de Lyon, 69495 Pierre-Bénite, France

**Keywords:** melanoma, biomechanics, stiffness, EMT-TFs, tetraspanin 8

## Abstract

**Simple Summary:**

We aimed to investigate the intrinsic biomechanical properties of melanoma cancer cells. We demonstrate that the modulation of stiffness properties, and their correlated morphometrics, impacts cutaneous melanoma cell invasiveness following EMT switching and Tspan8 transcriptional activation. Thus, we propose that melanoma stiffness measurement could be of prognostic relevance for patients.

**Abstract:**

The intrinsic biomechanical properties of cancer cells remain poorly understood. To decipher whether cell stiffness modulation could increase melanoma cells’ invasive capacity, we performed both in vitro and in vivo experiments exploring cell stiffness by atomic force microscopy (AFM). We correlated stiffness properties with cell morphology adaptation and the molecular mechanisms underlying epithelial-to-mesenchymal (EMT)-like phenotype switching. We found that melanoma cell stiffness reduction was systematically associated with the acquisition of invasive properties in cutaneous melanoma cell lines, human skin reconstructs, and Medaka fish developing spontaneous MAP-kinase-induced melanomas. We observed a systematic correlation of stiffness modulation with cell morphological changes towards mesenchymal characteristic gains. We accordingly found that inducing melanoma EMT switching by overexpressing the ZEB1 transcription factor, a major regulator of melanoma cell plasticity, was sufficient to decrease cell stiffness and transcriptionally induce tetraspanin-8-mediated dermal invasion. Moreover, ZEB1 expression correlated with Tspan8 expression in patient melanoma lesions. Our data suggest that intrinsic cell stiffness could be a highly relevant marker for human cutaneous melanoma development.

## 1. Introduction

Melanoma, which originates from skin melanocytes after transformation and abnormal proliferation, is the most aggressive cutaneous cancer due to its high propensity to metastasize. Despite prevention campaigns and the recent development of targeted and immune therapies, cutaneous melanoma remains the deadliest skin cancer, with a rapid increase in incidence over the past thirty years [1]. The main cause of death due to cutaneous melanoma is metastases dissemination and the capacity of melanoma cells to resist treatment. Indeed, cutaneous melanomas are very heterogeneous tumors composed of cells that are dynamically regulated, at the genetic, epigenetic, and phenotypic levels, by the selective pressures imposed by the host tumor microenvironment and the host immune system. Melanoma cells can thus switch between different transcriptional programs and differentiation states during melanoma progression [2,3,4,5,6]. They can also adapt to therapeutic pressure by inducing transcriptomic and epigenetic programs during the early treatment phases [7,8,9,10]. These programs generally lead to cell dedifferentiation, characterized by a decrease in cell proliferative signatures in favor of a gain in molecularly invasive ones. This melanoma cell plasticity is associated with “phenotype switching”. This “phenotype switching” is driven at the transcriptional level by an expression loss in the microphthalmia-associated transcription factor (MITF) and the SOX10, SNAI2, and ZEB2 genes and an expression increase in the ZEB1, AXL, and SOX9 genes [11,12]. Some of these genes (SNAI2, ZEB1, and ZEB2) that mediate cell plasticity belong to the family of transcription factors that regulate epithelial-to-mesenchymal transition (EMT). EMT is an essential embryonic process that provides a cell motility capacity and is frequently reactivated during carcinoma development [13,14]. In melanoma, which develops from melanocytes that are already presenting mesenchymal features, the expression of EMT transcription factors (EMT-TFs) is nevertheless crucial for melanoma progression [15]. Indeed, the loss of SNAI2/ZEB2 combined with the gain of ZEB1 is a factor involved in the poor prognosis of melanoma patients [16] and is associated with resistance to MAPK-targeting [17] and immune [18] therapies. Specifically, ZEB1 is a major driver of phenotype switching by transcriptionally regulating the switch from SOX10-MITF-dependent differentiation programs towards AP1-dependent invasive programs [19]. The cell plasticity and adaptative capacities of melanoma cells could explain why, even if the molecular mechanisms underlying cutaneous melanoma transformation and progression towards an invasive state are deciphered, the efficient targeting of melanoma cells remains difficult.

Thus, there is growing evidence that the study of melanoma progression mechanisms has to take into account the interplay between the tumor cells and their microenvironment, especially in a cancer type where cell plasticity plays such a central role. Research in recent decades has mainly focused on the role of the immune system in cancer initiation and progression and in resistance to therapy [20,21], but very few studies have explored how melanoma cells adapt their morphological and biomechanical features to their microenvironment during tumor progression [22]. However, for cutaneous melanoma cells, becoming more prone to escape enforced constraints in a highly stiff epidermal environment could be crucial for melanoma progression towards an invasive state. Some studies have focused on the biochemical modulation of the extracellular matrix (ECM) during melanoma development and showed that melanoma cells are indeed exposed to a more rigid microenvironment as the tumor grows [23,24,25]. Hirata et al. also showed that melanoma-associated fibroblasts in the ECM could provide a “safe haven” for melanoma cells, increasing the ECM rigidity following BRAF inhibitor treatment and thus protecting melanoma cells from targeted therapy [26]. It has also been proposed that a more aligned and stiffer ECM, as observed during aging, promotes melanoma cell invasion [27]. However, melanoma progression due to the acquisition of invasive properties by melanoma cells is probably a function of both ECM stiffness and the modulation of the intrinsic biomechanical properties of melanoma cells themselves. For instance, Weder et al. showed, by atomic force microscopy (AFM), decreasing stiffness from normal human melanocytes to locally invasive melanoma cell lines in vitro [28]. This suggests that melanoma cells, well known for their plastic abilities, could adapt their stiffness throughout tumor progression so as to respond optimally to the rigidification of their microenvironment.

In our study, we demonstrate that melanoma cell stiffness is a biomarker of melanoma progression in vitro, in vivo, and potentially in patients, linked to EMT-TF phenotype switching. We show that cell stiffness decreases during melanoma progression both in melanoma cell lines in 2D and in 3D human skin reconstructs (HSR), and also in the medaka fish melanoma model in vivo, a highly suitable model for melanoma study. We show that the reduction in stiffness is accompanied by melanoma cell morphological changes, which lead us to investigate the role of EMT-TFs in the melanoma cell morphology and stiffness modulation. We demonstrate that EMT-TFs’ expression switching, leading to Zeb1 overexpression, directly regulates the expression of the tetraspanin 8 (Tspan8) transmembrane protein and that Tspan8 expression is sufficient to remodel the melanoma cell morphology, decrease the intrinsic cell stiffness, and induce dermal invasion.

## 2. Materials and Methods

### 2.1. Cell Lines and Cell Culture

Non-invasive IC8 and invasive T1C3 melanoma clones previously described in [29] and WM115 cells (ATCC) were cultured as monolayers in McCoy’s 5A medium (Gibco, London, UK) supplemented with 10% fetal calf serum, 100 IU/mL penicillin, and 100 IU/mL streptomycin. Clones T1C3 and IC8 were generated by the limiting dilution cloning of M4Be cell lines [29,30]. The same conditions were applied to stable tspan8-positive and tspan8-negative clones generated with shRNA-mediated silencing or ectopic overexpression and described in [31,32]. A melanocyte growth medium kit was used for normal human melanocytes (NHM) (Promocell, Heidelberg, Germany). Patient-derived BRAFV600 melanoma cell lines (C-09.10) were kindly provided by Dr Robert Balloti (Nice, France) and cultured as monolayers in RPMI medium (Gibco, London, UK) supplemented with 7% fetal calf serum, 100 IU/mL penicillin, and 100 IU/mL streptomycin. All these cell lines were cultured under standard conditions and tested as mycoplasma-free, as previously described [29]. ZEB1-overexpressing C-09.10 cells were previously described [17].

### 2.2. Human Skin Reconstructs (HSR)

Adult human keratinocytes (4 × 10^5^ cells·cm^−2^), mixed with NHM or human melanoma cells (5.820 cells·cm^−2^) at a melanoma/keratinocyte (or melanocyte/keratinocyte) ratio of 1:80, were seeded into a stainless-steel ring deposited on the surface of human dead de-epidermized dermis (DED) squares, as previously described [33]. After 21 days of incubation at an air–liquid interface, the specimens were collected for atomic force microscopy analysis. HSR were measured as a whole or as cryosections, as described in the dedicated sections below.

### 2.3. Japanese Medaka (Oryzia latipes) Strains and Breeding

All medaka experiments were conducted with mitf::Xmrk/+ or mitf::Xmrk/+ p53-/- strains [34]. These fish develop spontaneous malignant melanoma due to the constitutive activation of the EGF receptor under the MITF promoter, which allows one to express this activation only in fish pigment cells (melanophores). The medaka melanoma shows the same histopathological features as in their human counterparts and they have matching expression profiles [35,36]. The general maintenance and collection of adult medaka fish were carried out at the PRECI facilities (IFR128, Biosciences Gerland, Lyon, France; agreement of the French Ministry n° B-69387-0602), as previously described [37].

### 2.4. Transient Transfections

Melanoma cells were seeded at 2 × 10^5^ cells per well in 6-well plates. For siRNA transfections, cells were transfected 24 h after seeding in transfection medium OPTIMEM (Gibco, UK) using 2 μL INTERFERin (Polyplus, Illkirch, France) and with 20 nM of control, SNAI2, or ZEB2 siRNA (Qiagen GmbH, Hilden, Germany) or with the ON-TARGET plus Smart Pool siRNA specific to TSPAN8 (Smart pool, Dharmacon, Chicago, IL, USA). Cells were collected 24 to 48 h after transfection for real-time RT-QPCR experiments, chromatin immunoprecipitation assays, and stiffness measurements, and 48 to 72 h after transfection for Western blotting experiments.

### 2.5. Real-Time RT-qPCR

Total RNA was extracted using the RNAeasy mini-kit (Qiagen GmbH, Hilden, Germany), reverse-transcribed into cDNA by the RT Maxima Reaction Kit (ThermoFisher, Waltham, MA, USA), and analyzed by real-time qPCR using the ONEGreen^®^ Fast qPCR Premix (Ozyme, Danvers, MA, USA) on an Azure CelioTM 3 Real-Time PCR System (Ozyme, Danvers, MA, USA). Results were obtained from at least three independent experiments and normalized to the GAPDH or ACTIN expression level for human and medaka experiments, respectively. The human primers used were as follows: SNAI2-F: 5′-AGG AAT CTG GCT GTG-3′; SNAI2-R: 5′-GGA GAA ATG CCT TTG GAC-3′; ZEB1-F: 5′-GAT GAT GAA TGC GAG TCA GAT GC-3′; ZEB1-R: 5′-ACA GCA GTG TCT TGT TGT A-3′; ZEB2-F: 5′-AAG CCA GGG ACA GAT CAG-3′; ZEB2-R: 5′-GCC ACA CTC TGT GCA TTT GA-3′; GAPDH-F: 5′-CCG GGA AAC TGT GGC GTG ATG G-3′; GAPDH-R: 5′-AGG TGG AGG AGT GGG TGT CGC TGT T-3′; TSPAN8-F: 5′-TTG CTT CTG ATC CTG CTC CT-3′; TSPAN8-R: 5′-AGG GCC TGC AGG TTC ACA CCA C-3′. The medaka primers used were as follows: SNAI2-F: 5′-TCA CAC GTT GCC TTG TGT TT-3′; SNAI2-R: 5′-TTG GAG CAG TTC TTG CAT TG-3′; ZEB1-F: 5′-CGA GTG TGG CAA AGC GTT TA-3′; ZEB1-R: 5′-TGC CTG CCG TTC ATT GAG AT-3′; ACTIN-F: 5′-AGT CCT GCG GTA TCC ATG AG-3′; ACTIN-R: 5′-AGC ACA GTG TTG GCG TAC AG-3′; TSPAN8-F: 5′-CTG TGG GAT CAT CCA AGG AC -3′; TSPAN8-R: 5′-CCA GCA CCA CCT TCA TGT TT-3′.

### 2.6. Protein Extraction and Western Blotting

Protein extraction and Western blotting were performed exactly as previously described [32]. The used antibodies were as follows: mouse monoclonal TS29 clone (1/500) for human Tspan8 detection; sc-166476 mouse antibody (1/500, Santa Cruz, CA, USA) for human SNAI2 detection; rabbit monoclonal antibody HPA027524 (RRID:AB_1844977, Sigma, Burbank, CA, USA) for human ZEB2 detection; rabbit monoclonal antibody HPA003456 (1/500, RRID: AB_10603840, Sigma, USA) for human ZEB1 detection; custom antibody elaborated from rabbit serum before and after immunization (Covalab, Bron, France) for Medaka Tspan8 detection; mouse monoclonal anti-actin clone C4 antibody (1/5000; MAB1501, Millipore, Darmstadt, Germany) for both human and medaka β-actin detection. The HRP-conjugated secondary antibodies were a mouse anti-rabbit IgG (1/2500, sc-2492, Santa Cruz, CA, USA) and a mouse IgG (1/2500, sc-2025, Santa Cruz, CA, USA). Western blot detections were performed using the Clarity Western ECL Substrate (Bio-Rad, Hercules, CA, USA). Digital Imaging was performed with the ChemiDoc MP Imager (Bio-Rad, Hercules, CA, USA). Western blot quantifications were achieved using the ImageJ software (1.8.0). At least two independent biological replicates were performed.

### 2.7. Chromatin Immunoprecipitation

ChIP experiments were carried out according to the protocol of the iDeal ChIP-Seq Kit for Transcription Factors (Diagenode, Denville, NJ, USA). Patient-derived cells from one 15 cm dish were cross-linked with 1% formaldehyde for 10 min at RT. They were then quenched for 5 min in 125 mM glycine. The cross-linked chromatin was isolated and sonicated in order to generate small DNA fragments (200–500 bp in length on average) using a Bioruptor plus sonication device (Diagenode, Denville, NJ, USA). Chromatin fragments were then immunoprecipitated with antibodies directed against Zeb1 (1 μg, GTX105278, Genetex) or IgG (1 μg, PRABP01, Bio-Rad, Hercules, CA) as a negative control. Immunoprecipitated DNA was purified and then resuspended in H2O (50 μL). DNA was finally analyzed by qPCR using the QuantiTect SYBR ^®^ Green PCR kit (Qiagen GmbH, Hilden, Germany). qPCRs were run on the Azure CelioTM 3 Real-Time PCR System (Ozyme, Danvers, MA, USA). Primers used were as follows: control-upstream-region-F: 5′-GGA ATT TCC AGG AGT GAA CTG-3′; control-upstream-region-R: 5′-ATT TTG GTG GAT GCT GAA CA-3′; MITF-F: 5′-CTG AAG ATC CCA GCG GGT TG-3′; MITF-R: 5′-GAG GTG ACT CCA AGC GAA CT-3′; pTSPAN8-F: 5′-TGA TAA CAG GTT GCT ATG TCT AAG C-3′; pTSPAN8-F: 5′-TTT TGC ATT GCA TTT TCT CAA-3′. Relative promoter enrichment was normalized to chromatin inputs.

### 2.8. Cryosections

Reconstructed human skin and medaka fish were cryopreserved in isopentane cooled to −80 °C by dry ice after being embedded in OCT (OCT Cellpath). Then, 30 µm sections were created with a Leica CM30505 cryostat on Superfrost plus slides (Thermo Scientific, Waltham, MA, USA).

### 2.9. Immunohistochemistry

Cryosections of reconstructed human skin were stained with Fontana Masson (ab150669) to reveal melanin pigments in black. Immunochemical analyses for Tspan8 staining were performed on 18 archival formalin-fixed paraffin-embedded melanoma tumor specimens from the Department of Biopathology of Lyon Sud Hospital (France), as previously described in [29]. Tspan8 staining of medaka sections was performed with the custom antibody elaborated from rabbit serum before and after immunization (Covalab, Bron, France).

### 2.10. Optical Microscopy and Fluorescence Acquisition

The images were taken with a Zeiss Observer z1 LSM 880 inverted microscope with a ×40 (Zeiss, Oberkochen, Germany) (1.4) plane Apochromat lens. For optical microscopy, melanoma cells were cultured on Ibidi slides (80841) and rinsed in 1X PBS before being fixed in 3.7% formaldehyde (Sigma 252549). The cells were then rinsed in PBS and cortical filamentous actin was stained with a solution of phalloidin (phalloidin-tetramethylrhodamine B isothiocyanate, Sigma P1951). DAPI (Sigma 28718-90-3) staining revealed the nucleus. The slides were mounted (slide/lamellar) in PBS 1X/glycerol (50/50).

### 2.11. Image Analysis

Image analysis via light microscopy was processed with the free software Fiji (NIH, ImageJ software (1.8.0)).

### 2.12. Atomic Force Microscopy (AFM)

The stiffness measurements were performed with a Resolve Bioscope (Bruker Nano Surface, Santa Barbara, CA, USA) mounted on an inverse microscope (DMi8, Leica, Wetzlar, Germany), equipped with a ×20–×40 air objective (Leica, Germany). The AFM acquisition software version was 9.1.

### 2.13. Atomic Force Microscopy Data Acquisition

A schematic describes how the stiffness was measured in the cell culture, skin reconstructs (whole skin and cross-sections), and medaka fish (whole skin and cross-sections; Appendix A).

For melanoma cells in cultures, indentation force measurements were performed with cantilevers with a spring constant of 0.4 N/m with a pyramidal tip geometry. The tip radius provided by the manufacturer was 8 to 12 nm (ScanAsyst-Air, Bruker AFM Probes, Inc., Camarillo, CA, USA). Each cantilever was calibrated using the thermal tune method and the deflection sensitivity was determined in contact mode on a sapphire. The Petri dishes containing the in-vitro-cultured cells in their medium were placed at the last moment on the motorized heating plate (xy) of the AFM. The AFM experiment consisted of using the Miro QNM mode in fluid by realizing arrays of 256 curves (FV) of indentation force on a range of 100 µm^2^ to 7 Hz using 2048 points per curve.

For human skin reconstructs (HSR), indentation force measurements were performed with cantilevers with a spring constant of 40 N/m with a spherical tip geometry. The tip radius provided by the manufacturer was 500 nm (Biosphere B500-NCH). Each lever was pre-calibrated (spring constant) and the deflection sensitivity was determined in contact mode on a sapphire. Each reconstructed human skin specimen was mounted on a support (BioMecaR design), which allowed uniform and constant tension on the whole sample. The support was then placed on the motorized stage (xy) of the AFM. The AFM experiment consisted of using the extended Z contact mode by realizing arrays of 256 indentation force curves over a range of 400 µm^2^ to 1 Hz using 2048 points per curve.

For cryosections of HSR and medaka, indentation force measurements were performed with 0.4 N/m spring constant cantilevers with a pyramidal tip geometry. The tip radius provided by the manufacturer was 8 to 12 nm (ScanAsyst-Air, Bruker AFM Probes, Inc.). Each lever was calibrated using the thermal tune method and the deflection sensitivity was determined in contact mode on a sapphire. The slide containing the cryosections were rehydrated for 30 min and then placed on the motorized stage (xy) of the AFM. The AFM experiment consisted of using the Miro QNM mode in fluid by realizing arrays of indentation force curves on a variable measurement range according to the area with an adapted resolution. The acquisition was performed at 7 Hz using 2048 points per curve.

### 2.14. Atomic Force Microscopy Data Analysis

The extraction of the global elastic modulus was performed by applying the theoretical model of Hertz (HSR) or Sneddon (cells and cryosections). The extraction of the force volumes was performed with the Nanoscope Analysis 3.0 software, and they were then analyzed with the BioMeca analysis software (1.3.7). The analyses were performed on an indentation of 2 µm on cultured cells, of 5 µm on reconstructed human skin, and of 700 nm on cryosections. The models used were intended for flat surface indentations by a rigid indenter but were accepted in our study. Furthermore, the indentation of less than one third of the sample ensured that only the desired structures were measured.

Sneddon model:$$F=\frac{2}{\pi }\;\cdot \;\frac{E}{1-\nu^{2}}\;\cdot \;tan(\alpha )\cdot \delta^{2}\;$$

Hertz model:$$F=\frac{4}{3}\;\cdot \;\frac{E}{1-\nu^{2}}\;\cdot \;\surd R\delta^{3/2}$$
where F is the force, E is the Young modulus, ν is the fish coefficient, *R* is the tip radius, and δ is the indentation.

Here, we chose 0.3 as the fish coefficient. The measurements obtained were expressed as a modulus, except Ea, which corresponded to a realistic approximation of the stiffness of the sample. All results were treated in the same way, which allowed us to work in relative terms.

### 2.15. Statistical Analysis

All data represented with error bars are the means and S.D or S.E.M. of at least three independent experiments. Statistical analysis was performed using the R/R studio software (3.3.0). The normality tests were performed with the Shapiro test, and the homoscedasticity tests with the Levene test. Depending on the results, Student’s *t*-test or Wilcoxon’s test was applied to compare the different means. The acceptability was set at 5%. Mean differences were considered significant when * *p* < 0.05, ** *p* < 0.01, *** *p* < 0.001; ns = not significant.

## 3. Results

### 3.1. Stiffness Properties Decrease during Melanoma Transformation and Progression

In order to confirm data from the literature, we first examined the stiffness in vitro in 2D culture cell lines by atomic force microscopy (AFM), a microscopy technique using a type of scanning probe microscope, particularly suitable for studying both adherent cells and tissues. Consistently with the literature data, we found that normal human primary melanocytes (NHM) exhibited higher stiffness values (around 4 kPa) compared to non-invasive tumoral melanoma cells and even more invasive tumoral melanoma cells (Appendix A). Specifically, we measured the stiffness differences between two cell clones derived from the same M4Be parental cell line and displaying different invasiveness capacities in vivo: the T1C3 clone is able to give high numbers of spontaneous metastases in the lungs of immunosuppressed newborn rats, whereas the IC8 clone is able to give only very few metastases in the same model [30]. In HSR, T1C3 cells are consistently able to invade the dermis, in contrast to their parental M4Be cell line and the IC8 clone. Measurements of the biomechanical properties by AFM revealed a significant two-times stiffness reduction in the very invasive T1C3 clone compared to its non-invasive IC8 counterpart (Appendix A). These data thus confirm that melanoma progression towards an invasive state is accompanied by a drastic intrinsic cell stiffness decrease in melanoma cell 2D cultures.

In order to study melanoma cells’ biomechanical properties in more physiological models, we first decided to explore stiffness in a 3D model of human skin reconstructed with healthy melanocytes (NHM) or with the previously described melanoma cells displaying various invasive properties (IC8 and T1C3 [38]). We thus measured the stiffness directly on the whole HSR and observed a significant decrease in global stiffness in human skin reconstructed with melanoma cells compared to human skin reconstructed with NHM. Moreover, we noticed a greater stiffness decrease when melanoma cells displayed an invasive phenotype (T1C3 cells; Figure 1a, left panel). To confirm that the global stiffness modulation in the HSR was essentially due to the status of melanocytes or melanoma cells, we then measured the biomechanical properties on HSR sections, focusing on melanoma areas observed by microscopy. Whereas the stiffness of human skin reconstructed with IC8 non-invasive melanoma cells was around 16,000 kPa, the stiffness of human skin reconstructed with T1C3 invasive melanoma cells significantly decreased until 12,000 KPa (Figure 1a, right panel). These data confirm in a 3D model that the acquisition of invasiveness by melanoma cells is accompanied by their intrinsic stiffness decrease.

Finally, to determine whether the stiffness reduction throughout melanoma acquisition and progression could also be a biomechanical feature measured in vivo, we used a transgenic mitf::Xmrk/+ medaka fish model. This fish strain overexpresses the oncogenic receptor tyrosine kinase Xmrk and spontaneously develops invasive melanomas sharing numerous properties with patient melanomas [34,39,40]. We previously set up AFM measurements on the whole medaka fish [37]. Thus, we analyzed stiffness from healthy skin versus melanoma lesions, and, since the basal stiffness could vary according to the localization of the lesion, we performed measurements both on dorsal areas and on lateral areas. We observed that, wherever the melanoma appeared, a systematic significant stiffness reduction occurred on the melanoma lesion compared to the surrounding healthy skin (Figure 1b, left panel). When a stiffness projection was applied on the measured area, we clearly distinguished softer regions (in blue) corresponding to pigmented melanoma lesions versus stiffer adjacent regions (in yellow to red) corresponding to healthy non-pigmented surrounding skin (Figure 1c). We confirmed the stiffness decrease during melanoma development in another medaka fish model (Figure 1b, right panel), in which the additional inactivation of the TP53 gene (mitf::Xmrk/+ p53-/-) led to the rapid development of melanomas with a strikingly larger size [34]. Finally, by grading the melanoma progression in medaka fish by determining the area covered by melanoma and measuring the associated stiffness values, we established that the stiffness was progressively reduced during melanoma progression (Figure 1d).

Overall, our data demonstrate that the stiffness was systematically and significantly decreased throughout melanoma progression towards an invasive state in relevant melanoma models.

### 3.2. Stiffness Properties Are Associated with Cell Morphological Changes and with Modulation of Expression of Transcription Factors Regulating Epithelial-to-Mesenchymal Transition

Since biomechanical properties are unquestionably correlated with cytoskeleton organization and the cell morphology, we analyzed different parameters of the cell shape in our melanoma models. The comparison between the IC8 non-invasive and T1C3 invasive melanoma clones, derived from the same parental melanoma cell line, allowed us to evaluate the morphological plasticity that could be acquired by melanoma cells simultaneously with invasiveness properties. The traditional cell shape descriptors are the aspect ratio (corresponding to base cell shape), area, solidity (corresponding to margin undulation), and circularity [41]. The results showed that the area and solidity parameters seemed not to be affected between the non-invasive and invasive melanoma cell lines, in contrast to the aspect ratio and circularity, which were, respectively, increased and decreased in T1C3 invasive cells (Figure 2a, left panels). These data perfectly correlate with microscopic observations showing that non-invasive melanoma IC8 cells are rounder than T1C3 invasive ones, which appeared more mesenchymal (Figure 2a, right panel). The analysis of sections from HRS showed that, as for 2D cultured cells, the area was not impacted but the cell aspect ratio and circularity tended to increase and decrease, respectively, with invasiveness, even if the values were not significantly different (Figure 2b). The cell morphology has been shown to be affected by the epithelial-to-mesenchymal transition (EMT) in various cancer cell types and this can be linked to the modification of the cell’s biomechanical properties. Even if the effect of the extracellular matrix on cell stiffness has been extensively studied in the past few years, it appears that the intrinsic cell properties in terms of morphometrics and biomechanics could also be tightly correlated with the EMT process, especially in cancer cells (for review, see [42]). For instance, TGF-β, known to play a key role in EMT, is able both to reorganize the cytoskeleton and to modify the rigidity and invasiveness in small-cell lung carcinoma (NSCLC) cells [43]. In head and neck cancer cell lines cultivated in a 3D matrix environment, the EMT status affected the cell morphology, stiffness, and invasiveness, and the knockdown of the EMT transcription factors (EMT-TFs) TWIST1 and SNAI1 was sufficient to modulate the intrinsic stiffness of these cancer cells [44]. In melanoma, we previously showed that, during melanoma progression, a switch in the EMT-TFs’ expression occurs, with a loss of ZEB2 and SNAI2 expression and the upregulation of ZEB1 and TWIST1 expression. This expression switch is associated with melanoma cell plasticity and represents a major risk factor for a poor outcome in melanoma patients [16], driving resistance to targeted [17] and immune [18] therapies.

Therefore, to determine whether EMT-TFs could directly impact both melanoma cell stiffness and cell shape, we analyzed, by AFM, melanoma cells in which the expression of EMT-TFs was modulated. To this end, we overexpressed ZEB1, leading to a decrease in E-cadherin expression and an increase in N-cadherin expression (Figure 2c), or silenced SNAI2 or ZEB2 in melanoma cells in 2D cultures. We observed that Zeb1’s stable overexpression in melanoma cells, known to induce invasive signatures [19], was sufficient to significantly decrease cell stiffness (Figure 2d, left panel). This reduction in cell stiffness was accompanied by changes in cell morphology, with a decrease in circularity and an increase in aspect ratio with ZEB1 overexpression (Figure 2d, right panels). These data show that modulating only ZEB1 expression was sufficient to drastically modify the cell shape and intrinsic cell stiffness. Similarly, the inhibition of SNAI2 or ZEB2 expression by siRNA, which led to an increase in ZEB1 expression (Figure 2e), resulted in a significant reduction in cell stiffness (Figure 2f, left panel) correlated with an increase in the aspect ratio of melanoma cells and a decrease in circularity (Figure 2f, right panels).

Overall, our results show that the modulation of EMT-TF expression, resulting in ZEB1 overexpression, which is crucial for melanoma progression, is sufficient to remodel the melanoma cell shape and alter their biomechanical properties.

### 3.3. Tetraspanin 8 (Tspan8) Is the Major Target of EMT-TFs for Cell Stiffness and Morphology Regulation during Melanoma Progression

We previously demonstrated that Tspan8 is a transmembrane protein sufficient to confer invasiveness to non-invasive melanoma cells both in 2D cultured cells and 3D skin reconstructed models [32,33,45,46,47]. We also performed an RNA-interference-based screen to identify Tspan8 transcriptional regulators whose deregulation could lead to the appearance of Tspan8 expression and the concomitant acquisition of invasive properties [32]. In this screen, we identified SNAI2 as a potential Tspan8 inhibitor. SNAI2 was the only EMT-TF tested in the genetic screen (focused on a “metastasis set” of genes), but we can hypothesize that the global process of EMT expression switching could affect Tspan8 expression. We therefore questioned whether Tspan8 could mediate EMT-TFs’ effect on the biomechanical and morphological properties of melanoma cells.

We first validated Tspan8 as a direct transcriptional target of the EMT-TF expression switch. We found that silencing the SNAI2 and ZEB2 transcription factors, which behave as tumor suppressors in melanoma cells, significantly induced Tspan8’s endogenous expression at both the mRNA (Figure 3a) and protein (Figure 3b) levels in different melanoma cell lines. Since we observed that SNAI2 or ZEB2 inhibition led, as expected during the phenotypic switch, to a significant increase in ZEB1 expression (Figure 2e), we thus tested whether ZEB1 modulation could also affect Tspan8 expression. We demonstrated that ZEB1 overexpression in melanoma cells was indeed able to activate Tspan8 mRNA (Figure 3c, upper panel) and protein (Figure 3c, lower panel) expression. A previous analysis of ZEB1’s direct transcriptional targets by chromatin immunoprecipitation (ChIP) experiments coupled with sequencing (ChIP-seq) identified a potential binding site in the TSPAN8 promoter (pTSPAN8) in pancreatic cells [48]. We thus hypothesized that ZEB1 could be recruited on pTSPAN8 in melanoma cells. By examining Ebox (5′-CANNTG-3′) presence in the TSPAN8 promoter region, we found such a putative ZEB1 binding site at position +10 downstream of the TSS. We therefore performed ChIP experiments using a ZEB-1 antibody versus a control immunoglobulin and used quantitative PCR to analyze ZEB1 enrichment on pTSPAN8. We found that, compared with a negative control located −1 kb upstream of pTSPAN8, ZEB1 was enriched two-fold on the positive control MITF (as shown in [19]) and enriched three-fold on the ZEB1 Ebox consensus site of pTSPAN8 (Figure 3d). These results indicate that endogenous ZEB1 is specifically recruited onto the Ebox site of the TSPAN8 promoter in melanoma cells.

Interestingly, we found that the switch in EMT-TF expression that we described in human melanomas [16] was also clearly observed during melanoma progression in medaka fish and was associated with an increase in Tspan8 expression. We collected samples from mitf::Xmrk/+ medaka and compared SNAI2, ZEB1, and Tspan8 expression in normal skin, primary melanomas, and melanoma local metastases (Figure 3e). We observed a significant decrease in SNAI2 mRNA expression correlated with an increase in ZEB1 mRNA and protein expression (Figure 3e, upper panels) along with melanoma progression towards an invasive state. In melanoma fish samples, we detected the appearance of Tspan8 expression both at the mRNA (Figure 3e, upper left panel) and at the protein level, detected by Western blot (Figure 3e, upper right panel) and immunohistochemistry (Figure 3e, lower panel).

Finally, we evaluated the Tspan8 expression level in patient melanomas by immunohistochemistry according to the ZEB1 expression level, between ZEB1^high^, ZEB1^int^, and ZEB1^low^ melanomas, in a cohort of 18 patient samples. The classification of this cohort into three subgroups was previously established based on the intensity and percentage of cells expressing ZEB1 [17]. We obtained the corresponding Tspan8 expression levels by staining serial sections. We identified a positive correlation between ZEB1 and the Tspan8 expression level since ZEB1^high^ melanomas were mainly associated with high expression levels of Tspan8 (five out of seven), contrary to ZEB1^low^ melanomas, which showed low to no Tspan8 expression (Figure 3f, left panel). We also found a positive correlation in three out of five ZEB1^int^ melanomas, where ZEB1-positive areas were tightly associated with high corresponding Tspan8 expression areas (Figure 3f, right panel). Overall, our results suggest that the regulation of Tspan8 expression by EMT-TF expression switching is a crucial conserved mechanism for melanoma progression.

In order to determine whether Tspan8 could be a major downstream effector of the EMT-TF expression switch for biomechanical adaptation in melanoma cells, we first analyzed whether the modulation of Tspan8 expression could affect melanoma cell stiffness. Interestingly, in 2D cultured cell lines, we observed that cell stiffness was indeed modulated by Tspan8 expression. We previously generated different cell clones with various levels of Tspan8 expression: IC8 non-invasive cells stably transfected with a Tspan8 cDNA vector or an empty vector, and T1C3 invasive cells stably transfected with an shRNA targeting Tspan8 (T1C3/shTspan8) or a control shRNA (T1C3/shcontrol) [32]. We found that ectopic Tspan8 expression (Figure 4a, left panel) or the enrichment of Tspan8 expression by FACS sorting (Figure 4a, middle panel) strongly reduced melanoma cell stiffness, whereas Tspan8’s stable inhibition by shRNA increased their stiffness (Figure 4a, right panel). As expected, the morphological parameters of melanoma cells in which Tspan8 expression was modulated were also consequently modified (Figure 4b). We also found that ZEB1’s effect on the biomechanical properties of melanoma cells was dependent on Tspan8’s function, since the significant stiffness decrease induced by ZEB1 overexpression was abolished upon Tspan8 knockdown (Figure 4c).

We then confirmed the central role of Tspan8 in stiffness modulation in 3D HSR. We found on both the whole HSR (Figure 4d, left panel) and HSR sections (Figure 4d, right panel) that Tspan8’s ectopic expression was sufficient to decrease melanoma cell stiffness, whereas Tspan8’s inhibition increased it, confirming that Tspan8 expression could enhance melanoma dermal invasion by reducing melanoma cell stiffness.

In conclusion, we thus demonstrate that melanoma cell phenotype switching driven by the EMT-TF–tetraspanin8 axis is sufficient to decrease cell stiffness and promote dermal invasion.

## 4. Discussion

In our study, we demonstrate that cell stiffness reduction could be a biomarker of melanoma progression in vitro, in vivo, and potentially in patients harboring cutaneous melanoma. Very few studies have been performed to evaluate the potential of stiffness properties as a diagnostic and/or prognostic biomarker. It has been shown that, in mammary cancer cells, softer cells at the periphery of the tumor could facilitate tumor invasion and that eliminating the softer peripheral cells in mammary cancer organoids delays the transition towards an invasive phenotype [49]. These data perfectly correlate with ex vivo studies demonstrating that metastatic cancer cells from pleural effusions of patients suffering from lung, breast, and pancreatic cancers were 70% to 80% softer than their benign counterparts [50,51,52]. Moreover, a deformability cytometry microfluidic approach on the pleural effusions of patient samples clearly demonstrated that the softer and more deformable cancer cells are, the more malignant they are, for many different types of cancers [53]. Nanomechanical diagnosis could thus be an important tool to improve the early detection of cancer. However, limitations have been postulated for melanoma, especially since melanin could potentially interfere with the intrinsic mechanical cell properties. Indeed, the nanomechanical features of melanoma cells evaluated by AFM seem to show a correlation between the level of cell pigmentation and cell stiffness: non-pigmented cells could present a lower Young’s modulus and higher in vivo metastatic abilities than pigmented melanoma cells [54,55,56]. However, in our study, melanin did not seem to impact the melanoma cell stiffness or invasiveness, since, for instance, the T1C3 invasive cell line was more pigmented than the IC8 non-invasive one. Moreover, the melanomas that we analyzed in HSR or in vivo in medaka fish were pigmented melanomas with decreased stiffness despite pigment accumulation. Using stiffness as a potential diagnostic marker in patients would nevertheless take into account the pigmentation status of cutaneous melanoma, since we analyzed, in our study, only pigmented cutaneous melanomas. Indeed, amelanotic melanoma could be associated with poorer patient survival than pigmented melanoma [57].

Stiffness properties, however, cannot be used alone for diagnosis/prognosis and it is crucial to understand the molecular mechanisms underlying the adaptation of cancer cells in terms of biomechanical properties throughout progression towards an invasive state. Our study demonstrates that the intrinsic modulation of cell stiffness is importantly sustained by the transcription factors driving the pseudo-epithelial-to-mesenchymal transition (EMT-TFs) in melanoma. These transcription factors, and ZEB1 especially, have been previously shown by our laboratory as playing a central role in melanoma cell plasticity acquisition and driving both outcomes and therapy resistance in cutaneous melanoma patients [16,17,18]. It appears, in our study, that merely modulating EMT-TF expression is sufficient to remodel the melanoma cell shape and alter their biomechanical properties to favor invasiveness. The morphological and mechanical adaptation of melanoma cells induced by the EMT-TF expression switch seems to be mainly mediated by Tspan8 expression regulation. Indeed, Tspan8 modulation is sufficient to adapt both cell stiffness features and invasiveness, and ZEB1’s effect on the biomechanical properties of melanoma cells was dependent on Tspan8’s function, since the significant stiffness decrease induced by ZEB1 overexpression was abolished by Tspan8 inhibition. We previously demonstrated that Tspan8, a member of the tetraspanin family known to exert pro-invasive functions in numerous carcinomas, is sufficient to confer invasive properties to non-invasive melanoma cells [29,32,33,45,58] and could predict metastatic risk and patient outcomes in cutaneous melanoma [59]. Thus, we can propose that the combination of the immunohistochemistry staining of ZEB1 and Tspan8, whose expression is perfectly correlated in cutaneous melanoma patients’ lesions, with stiffness measurement in patients could be a new means to evaluate the dissemination potential of cutaneous melanomas.

Although the importance of the microenvironment has been extensively explored in recent years in terms of the immune response and escape, biomechanical properties have been studied to a lesser extent. However, the crosstalk between the immune and biomechanical responses could be strikingly correlated. Indeed, we previously showed that ZEB1 expression in melanoma cells was associated with decreased CD8+ T cell infiltration in melanoma tumors [18]. ZEB1’s ectopic expression is thus sufficient to repress the secretion of T-cell-attracting chemokines, leading to the impairment of CD8+ T cell recruitment and melanoma immune evasion. Moreover, it has been demonstrated that cell mechanical softness is a fundamental mechanism for cancer cells to evade T cell killing. Liu et al. showed that B16 melanoma mouse cells presenting stiffness <0.3 KPa, considered as “soft” cells, represent a subpopulation of melanoma cells that is able to survive to cytotoxic CD8+ T lymphocytes by using a mechanism preventing perforin from drilling a pore, which in turn leads to T cell killing evasion [60]. In particular, metastatic cells undergo F-actin remodeling, coupled with morphological and biophysical adaptations, which leads to a mechanotransduction response with the activation of myocardin-related transcription factors (MRTF) A and B [61]. MRTF overexpression has been shown recently to sensitize metastatic melanoma cells in vivo in mice to the immune system, establishing correlations between the biomechanical properties of melanoma cells and the capacity of cancer cells to activate and respond to cytotoxic lymphocytes [62]. In parallel with MRTF activation, YAP-TAZ activity, tightly coupled with the actin cytoskeleton architecture, has also been implicated in both the modulation of biomechanical features and BRAF inhibitor resistance [63]. Moreover, YAP expression in melanoma is directly correlated to PD-L1 expression, increasing the suppression of the activity of the cytotoxic CD8+ T lymphocytes, as well as decreasing cytokine production [64]. Since Tspan8 has also been shown to be a promising therapeutic target [65,66,67,68], especially for targeted radio-immunotherapy [69,70], we can consider a synergistic effect of Tspan8 blockade and softness inhibition. Beyond simply a biomarker for prognosis, biomechanical features could thus be considered as potential therapeutic targets in combination with currently used BRAF inhibitors or anti-PD1/anti-CTLA4 immunotherapies or Tspan8-blocking antibodies. Although actin-cytoskeleton-stabilizing agents, such as jasplakinolide, have been tested in vitro in melanoma cells [71], new therapeutical strategies based on stiffness enhancer (s) could be taken into consideration in light of biomechanical modulation.

## 5. Conclusions and Future Directions

This study allowed us to propose that cell stiffness could be a biomarker of melanoma progression in vitro, in vivo, and potentially in patients, linked to EMT-TF phenotype switching and Tspan8 expression regulation. Preliminary studies performed on biopsy explants from patients harboring melanoma lesions gave promising results and encourage us to further explore this mechanism. We also hope to discover the mechanism associated with TSPAN8-mediated stiffness changes and explore its potential mechanosensory role together with its partner, β1-integrin [31].

## Figures and Tables

**Figure 1 cancers-16-00694-f001:**
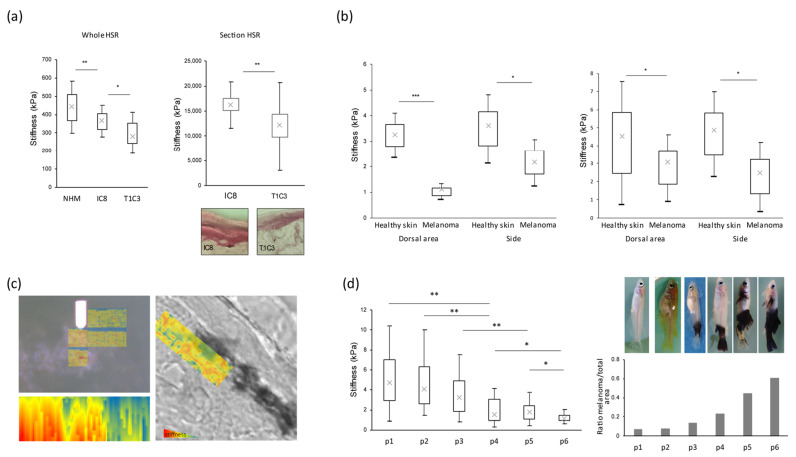
Melanoma transformation and progression are associated with stiffness decrease. (**a**) (**Top left panel**), overall stiffness measurement of whole human skin reconstructs (HSR) containing primary melanocytes (NHM) or melanoma cells. (**Top right panel**), stiffness measurement of melanoma cells in HSR cryosections. (**Bottom panels**), imaging of the corresponding HSR sections. (**b**) Global measurement of skin stiffness of medaka fish mitf::Xmrk/+ (**left panel**) or mitf::Xmrk/+ p53-/- (**right panel**) on healthy areas compared to areas with melanoma. Stiffness measurements were conducted on the flank or in dorsal areas of medakas. (**c**) (**Upper left panel**), optical and mechanical correlative images showing the rigidity of the skin at the interface between healthy area (white area (optical)) and melanoma area (black area (optical)) of the whole mitf::Xmrk/+ medaka. (**Bottom left panel**), tomographic reconstruction of the area scanned by AFM (projection in xz). (**Right panel**), optical and mechanical correlative image showing the stiffness of the skin at the interface between healthy and tumoral areas of cryosections from mitf::Xmrk/+ medaka. Blue corresponds to the lowest stiffness and red to the highest. (**d**) (**Left panel**), graphical representation of the stiffness of mitf::Xmrk/+ medaka (on tumoral areas) during melanoma progression. (**Upper right panel**), pictures of mitf::Xmrk/+ medaka with, respectively, from left to right, preneoplastic naevus to advanced invasive melanoma. (**Bottom right panel**), representative graph of the ratio between tumoral and healthy areas of mitf::Xmrk/+ medaka at different stages of tumor progression. Statistical significance of stiffness measurements was assessed by two-tailed Student’s *t*-test or Wilcoxon test, depending on the normality of the paired samples. Mean differences were considered significant when *p* < 0.05 (* *p* < 0.05; ** *p* < 0.01; *** *p* < 0.001).

**Figure 2 cancers-16-00694-f002:**
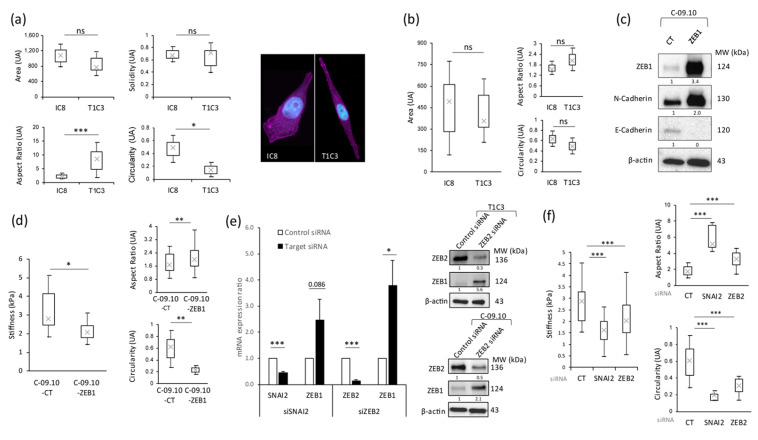
EMT-TF modulation regulates melanoma cell stiffness and morphological properties. (**a**) (**Left panel**), morphological analysis (area, solidity, aspect ratio, and circularity) of non-invasive (IC8) versus invasive (T1C3) melanoma cells in 2D culture. (**Right panel**), images of IC8 and T1C3 cells by confocal microscopy with the staining of actin (magenta) and nucleus (cyan). (**b**) Cellular morphological analysis (area, aspect ratio, and circularity) on human skin reconstructs (HSR) cryosection with non-invasive (IC8) and invasive (T1C3) melanoma cells. (**c**) ZEB1, N-cadherin, and E-cadherin expression measured by Western blot analysis in C-09.10 melanoma cells ectopically overexpressing ZEB1 (C-09.10-ZEB1) compared to the control condition (C-09.10-CT). (**d**) (**Left panel**), global stiffness measurement of C-09.10-CT versus C-09.10-ZEB1 melanoma cells. (**Right panel**), morphological analysis (aspect ratio and circularity) of C-09.10-CT and C-09.10-ZEB1 melanoma cells. (**e**) (**Left panel**), qPCR analysis of SNAI2, ZEB2, or ZEB1 transcript expression levels 72 h after SNAI2 (**left graph**), ZEB2 (**right graph**), or control (**both graphs**) siRNA transfection. (**Right panel**), ZEB2 and ZEB1 expression measured by Western blot analysis in T1C3 (**upper right panel**) or C-09.10 (**lower right panel**) melanoma cells 72 h after ZEB2 or control siRNA transfection. (**f**) (**Left panel**), global stiffness measurement of C-09.10 melanoma cells after control, SNAI2, or ZEB2 siRNA transfection. (**Right panel**), morphological analysis (aspect ratio and circularity) of C-09.10 melanoma cells after control, SNAI2, or ZEB2 siRNA transfection. GAPDH was used as a housekeeping gene in qPCR analysis, where data are shown as the mean ± SEM of three independent experiments, and β-actin was used as a loading control in Western blot analysis, where results are representative of two independent experiments. Statistical significance was assessed by two-tailed Student’s *t*-test or Wilcoxon test, depending on the normality of the paired samples. Mean differences were considered significant when *p* < 0.05 (* *p* < 0.05; ** *p* < 0.01; *** *p* < 0.001; ns non-significant). The original western blot figures can be found in Appendix A.

**Figure 3 cancers-16-00694-f003:**
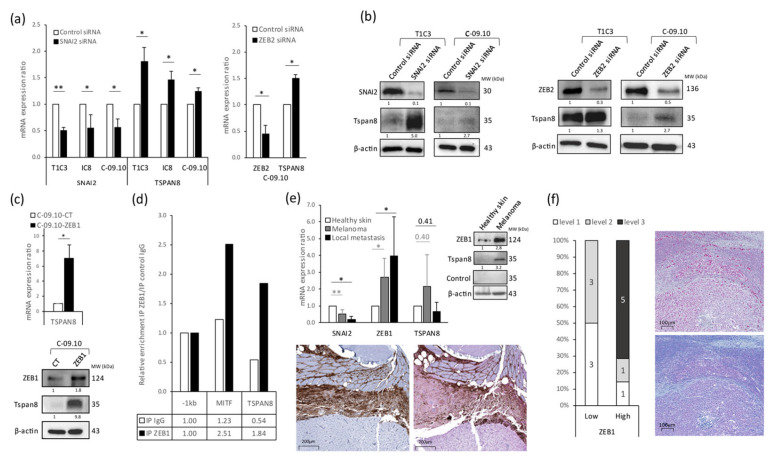
EMT-TFs regulate the expression of Tspan8, crucial for cutaneous melanoma invasion. (**a**) qPCR analysis of SNAI2, ZEB2, or TSPAN8 transcript expression levels 72 h after SNAI2 (**left panel**), ZEB2 (**right panel**), or control (**both panels**) siRNA transfection in non-invasive (IC8) or invasive (T1C3 and C-09.10) melanoma cells. (**b**) SNAI2, ZEB2, and Tspan8 expression measured by Western blot analysis 72 h after SNAI2 (**left panel**), ZEB2 (**right panel**), or control (**both panels**) siRNA transfection in T1C3 or C-09.10 melanoma cells. (**c**) (**Top panel**), qPCR analysis of TSPAN8 transcript expression level in C-09.10-ZEB1 compared to C-09.10-CT melanoma cells. (**Bottom panel**), ZEB1 and Tspan8 expression measured by Western blot analysis in C-09.10-CT and C-09.10-ZEB1 melanoma cells. (**d**) ZEB1 chromatin immunoprecipitation (ChIP) assays performed in C-09.10 melanoma cells, using IgG antibody as a negative control. Enrichment of TSPAN8 promoter region was analyzed by qPCR in comparison with a negative control promoter region located –1 kb upstream of the beginning of pTSPAN8 or a positive control using MITF promoter region. Results are representative of three independent experiments. (**e**) (**Upper left panel**), qPCR analysis of SNAI2, ZEB1, or TSPAN8 transcript expression levels in healthy skin, melanoma, or local metastasis conditions of mitf::Xmrk/+ medaka. (**Upper right panel**), ZEB1 and Tspan8 expression measured by Western blot analysis in healthy skin versus melanoma conditions of mitf::Xmrk/+ medaka. (**Lower panel**), Tspan8 staining in areas of invasive melanoma by immunohistochemistry on melanoma sections from a mitf::Xmrk/+ medaka fish, using a control antibody (serum before immunization, (**left image**) or a custom antibody directed against Tspan8 (serum after immunization, (**right image**). (**f**) (**Left panel**), quantification of Tspan8 expression level, based on the immunoscore previously established (47), in ZEB1^low^ (*n* = 6) and ZEB1^high^ (*n* = 7) melanoma samples. (**Right panel**), ZEB1 (**top**) and Tspan8 (**bottom**) expression analyzed by immunohistochemistry in ZEB1^int^ melanoma samples (*n* = 5). GAPDH was used as a housekeeping gene in qPCR analysis, where data are shown as the mean ± SEM of three independent experiments, and β-actin was used as a loading control in Western blot analysis, where results are representative of two independent experiments. Statistical significance of qPCR data was assessed by two-tailed Student’s *t*-test for paired samples, where mean differences were considered significant when *p* < 0.05 (* *p* < 0.05; ** *p* < 0.01). The original western blot figures can be found in Appendix A.

**Figure 4 cancers-16-00694-f004:**
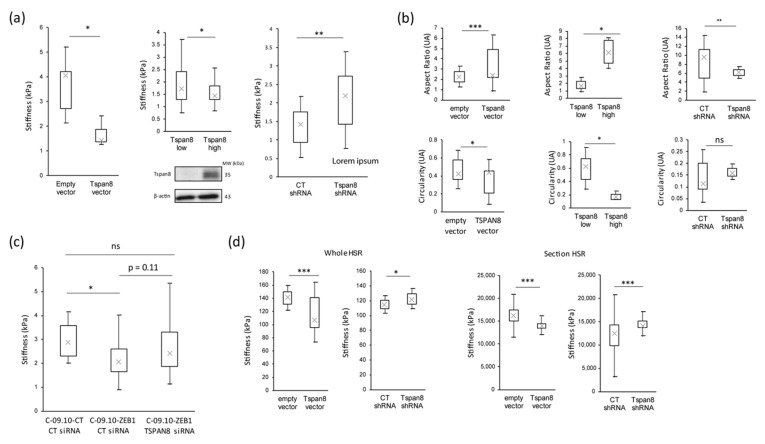
Tspan8 modulation regulates stiffness and morphological properties of cutaneous melanoma cells. (**a**) Global stiffness measurements of different melanoma cell constructs according to their Tspan8 expression levels measured by Western blot analysis (**bottom panels**). (**Top left panel**), non-invasive melanoma cells with an empty vector versus melanoma cells with a vector allowing ectopic expression of Tspan8. (**Top middle panel**), melanoma cells expressing very low level of Tspan8 versus the enrichment of the same cells expressing Tspan8. (**Top right panel**), melanoma cells strongly expressing Tspan8 with a control shRNA compared to the same cells transfected with a shRNA directed against TSPAN8. (**b**) Morphological analysis (aspect ratio and circularity) of melanoma cells according to their Tspan8 expression profile in correlation with (**a**). (**c**) Global stiffness measurement of melanoma cells expressing or not expressing ZEB1 and Tspan8 according to the ectopic expression of ZEB1 and transfection of control or TSPAN8 siRNA. (**d**) Global stiffness measurement of melanoma cells expressing or not expressing Tspan8 in human skin reconstructs (HSR). (**Left panel**), measurement at the surface of whole HSR with cells not expressing Tspan8 and cells ectopically expressing Tspan8 (**left**), along with cells expressing Tspan8 against the same cells expressing a shRNA against Tspan8 (**right**). (**Right panel**), measurement of melanoma cells on cryosections of HSR with cells not expressing Tspan8 and cells ectopically expressing Tspan8 (**left**) or cells expressing Tspan8 against the same cells expressing a shRNA against Tspan8 (**right**). β-actin was used as a loading control in Western blot analysis, where results are representative of two independent experiments. Statistical significance was assessed by two-tailed Student’s *t*-test or Wilcoxon test, depending on the normality of the paired samples. Mean differences were considered significant when *p* < 0.05 (* *p* < 0.05; ** *p* < 0.01; *** *p* < 0.001; ns non-significant). The original western blot figures can be found in Appendix A.

## Data Availability

Data generated or analyzed during this study are included in this published article or are available from the corresponding author on reasonable request, except for raw data from AFM measurements that were generated at BIOMECA. The derived data supporting the findings of this study are available from the corresponding author on reasonable request.

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
