# Peer review of "Cancer Cell Biomechanical Properties Accompany Tspan8-Dependent Cutaneous Melanoma Invasion"

_cancers, 2024, doi:10.3390/cancers16040694_

Round 1

Reviewer 1 Report

Comments and Suggestions for Authors

The authors observed a systematic correlation of stiffness modulation with cell morphological changes towards mesenchymal characteristics in melanoma. They demonstrated that inducing melanoma EMT-switching by overexpressing ZEB1 transcription factor, is sufficient to decrease cell stiffness and induce Tetraspanin8-mediated dermal invasion. It is a nice study with all apropriate controls. Would be nice to follow up on this analysis and measure changes in stiffness also upon mesenchymal to amoeboid transition.

Reviewer 2 Report

Comments and Suggestions for Authors

Cancers-2804031

Cancer cell biomechanical properties drive Tspan8-dependent cutaneous melanoma invasion

Runel et. al.

This manuscript highlights how melanoma stiffness is associated with disease aggressiveness. The group uses a mix of in vitro, 3D, and in vivo fish models to illustrate the implications of stiffness in melanoma progression. They uncover how the EMT related transcription factor Zeb1 positively regulates TSPAN8 – a 4x transmembrane containing protein – and TSPAN8 expression dictates melanoma stiffness. While a more robust analysis with panels of human melanoma cells would benefit the manuscript, overall this is an interesting article that provides insights to the physical nature of melanoma cells as a prognostic determinant. Specific points are outlined below:

Major

·         The title of this manuscript odd. The authors provide data illustrating how Tspan8 modulation regulates biomechanical properties (stiffness); not the other way around.

·         Figure 2a is may suffer from overinterpretation. The very elongated image of the TIC3 cells looks as though these cells are experiencing cell culture associated stress. This notion is potentially supported by the results of 2b. Have the authors run this analysis with a panel of other human melanoma cell lines? Of particular interest would be comparisons of patient matched cell line pairs. Ie. WM115 (VGP) to WM239A (Met).  

·         The stiffness measurements of the skin reconstructs and fish models are not well defined – The authors may consider a schematic indicting where/how the stiffness is measured.  Is the pressure measurement from the “top” of the skin reconstruct (the cornified layer) down through the dermal/epidermal junction OR are measurements taken via cross-sections? Is this the difference between the “whole HSR” and “section HSR”, respectively?.  If so, is the depth of the cornified layer taken into account when measuring from the skin surface down? The presence of an invasive melanoma would likely disrupt the cornified layer more so than an in situ melanoma.

·         TSPAN8 is known to exist in lipid rafts with elevated cholesterol levels. Does the ZEB1/TSPAN8 axis increase membrane cholesterol levels and thus cholesterol modulation is ultimately responsible for changes in cellular stiffness?

Minor

·         Perhaps it is a function of proofing and the available PDF, but a lot of the text in the figures is very small, with some being a washed out greyscale color.

·         Throughout the manuscript there are long-winded sentences.

·         What is the rationale for the Xmrk model?

Comments on the Quality of English Language

A number of long-winded sentences throughout this manuscript.

Reviewer 3 Report

Comments and Suggestions for Authors

The authors present in a concrete and precise way the possible importance of cell stiffness in melanoma progression. While the data presentation is clear there are some issues that have to be addressed:

1. There is no „conclusion“ section, which is mandatory in this journal.

2. In the methods section:

2.1. please clarify how the C-09.10, T1C3 and IC8 cell lines were obtained. It is not clear whether researchers from this paper got them, bought them or repeated the process of transfection for IC8 and T1C3

2.2. For transient transfection it would be suitable to note how many hours from transfection was passed when samples were collected

2.3. In section for protein extraction and western blotting, anti-tetraspanin antibody used in study was TS29 which is accompanied with reference. In the study that was referenced researchers did not use that antibody. Please remove the reference.

3. In the results section:

3.1 I suggest to put results on  2D culture stiffness in the supplement, as it was already described in literature, to have more space for properties in 3D culture. Figures are of poor quality and it was hard to follow all graphs. Results in HSR and Medaka fish model are described well, but figures are of poor resolution.

3.2 There is no shown western blot results for silencing SNAI2 and ZEB2 leading to increase of N-cadherin and decrease in E-cadherin. Western blots results should be provided in the supplemetal material. Same for the silencing SNAI2 leading in incerased expression of ZEB1, western blots shoud be provided in supplement.

3.3. In figure 3A in the graph on the right, it has to be clarified on which cell line the silencing the ZEB2 gene was performed.

4. I’m not sure if the title is appropriate, maybe it should be better to use „accompany“ rather than „drive“ Tspan8-dependent cutaneous melanoma invasion. 

Discussion is well written and all relevant findings were discussed. None of the results are overemphasized and authors’ statements were clear.

Round 2

Reviewer 2 Report

Comments and Suggestions for Authors

The authors have improved the manuscript however, a few points of concern remain:  

1.) Authors should at least speculate on a potential mechanism associated with TSPAN8 mediated stiffness changes. 

2.) As noted by other reviewers, the figure quality is still very poor and needs to be improved.
